# Persistent CD8 T Cell Marks Caused by the HCMV Infection in Seropositive Adults: Prevalence of HLA-E-Reactive CD8 T Cells

**DOI:** 10.3390/cells12060889

**Published:** 2023-03-14

**Authors:** Amélie Rousselière, Béatrice Charreau

**Affiliations:** 1Centre de Recherche Translationnelle en Transplantation et Immunologie (CR2TI), Nantes Université, CHU Nantes, Inserm, UMR 1064, 44093 Nantes, France; 2CHU Nantes, Institut de Transplantation Urologie Néphrologie (ITUN), CEDEX 1, 44093 Nantes, France

**Keywords:** HCMV infection, CD8 T cells, HLA-E, UL40, HCMV immunity

## Abstract

This study investigated the frequency and peptide specificity of long-lasting HCMV-specific CD8 T cells in a cohort of 120 cytomegalovirus seropositive (HCMV+) healthy carriers with the aim of deciphering the relative contribution of unconventional HLA-E- versus conventional HLA-A2-specific CD8 T cells to long-term T cell memory expansion in HCMV immunity. The presence of HCMV-specific CD8 T cells was investigated by flow cytometry using five MHC/peptide tetramer complexes (HLA-A2/pp65, HLA-A2/IE1 and three different HLA-E/UL40). Here, we report that 50% of HCMV+ healthy individuals possess HCMV-specific CD8 T cells, representing ≥0.1% of total blood CD8 T cells years post-infection. Around a third (30.8%) of individuals possess HLA-A2-restricted (A2pp65 or A2IE1) and an equal proportion (27.5%) possess an HLA-E/UL40 CD8 T response. Concomitant HLA-E- and HLA-A2-reactive CD8 T cells were frequently found, and VMAPRTLIL peptide was the major target. The frequency of HLA-E/VMAPRTLIL among total blood CD8 T cells was significantly higher than the frequency of HLA-A2pp65 T cells (mean values: 5.9% versus 2.3%, *p* = 0.0354). HLA-EUL40 CD8 T cells display lower TCR avidity but similar levels of CD3 and CD8 coreceptors. In conclusion, HLA-E-restricted CD8 T cells against the VMAPRTLIL UL40 peptide constitute a predominant subset among long-lasting anti-HCMV CD8 T cells.

## 1. Introduction

Human cytomegalovirus (HCMV) is a highly prevalent β-herpesvirus (namely HHV-5) that infects 60–90% of the world population. HCMV infection drives differential disorders according to the immunological status of the host. In immunocompromised individuals, including both solid organ and hematopoietic stem cell transplant recipients under immunosuppressive regimens and HIV-infected individuals, HCMV infection is a major risk factor for complications [1,2,3]. Congenital HCMV infection in immunologically immature newborns can cause severe morbidity, lifelong invalidity, and even mortality [4,5]. In healthy individuals, HCMV usually induces few or no clinical symptoms [6,7]. After a primary HCMV infection, the virus persists lifelong in a latent state in different cell types and under the control of the immune system.

HCMV immune control is multifactorial and there is evidence that B cells, CD8+ and CD4+ T cells, NK and γδT cells and antibody-mediated immunity all play a role in containment [8,9,10]. HCMV has a unique ability to evade host immune responses while simultaneously altering the homeostasis of host’s immune system. Memory inflation is a hallmark of HCMV infection and is characterized by an age-related increase in antigen-specific cells responding to defined immunodominant viral epitopes that maintain a terminally differentiated CD8^+^ T (CD8 T) cell phenotype over prolonged periods of time [9,11]. As a consequence of inducing memory inflation, HCMV drives oligoclonal expansions of T cell populations. Such a dramatic accumulation of virus-specific effector cytotoxic T lymphocytes (CTL) might impair the ability to respond to heterologous infection and may underlie the negative influence of HCMV seropositivity on survival in the elderly [12]. Some of these changes have been found to be particularly pronounced with aging, raising the possibility that HCMV may actively contributes to immunosenescence [13,14]. Moreover, evidence from large population studies points to an association of HCMV seropositivity with increased morbidity and mortality from cardiovascular diseases [15,16] and cancer [17].

CTL play an important role in the control of HCMV in the healthy host. Previous studies initially identified peripheral blood CD8+ CTL specific for the HCMV major immediate–early gene product (IE1, UL123) and for a structural gene product, the lower matrix protein pp65 (UL83) as predominant HCMV-specific CTL [18,19], leading to the assumption that these responses are immunodominant and representative of the total response to HCMV. However, CD8^+^ T cell recognition is broader than anticipated [20,21]. The pattern of recognized antigens and the response hierarchies (in terms of relative strength and frequency) are still not well understood, and actual correlates of immune protection are not known.

Major histocompatibility complex (MHC) class I allelic diversity is another trigger in the complexity of HCMV immunity. In addition to antigen-specific immunity orchestrated by MHC class Ia molecules, the non-classical MHC class Ib HLA-E molecules also mediate both innate [22,23] and adaptive immune responses [24,25]. Hoare et al. initially demonstrated the structural basis by which HLA-E mediates an adaptive MHC-restricted CTL response to UL40 peptides from HCMV [26]. They showed fine specificity for position 8 of the UL40 peptide, which discriminates self from non-self. The structure of the TCR-HLA-E complex is very similar to that of conventional TCR-MHC class Ia complexes. These results emphasize the evolutionary ‘duality’ of HLA-E, which not only interacts with the innate CD94/NKG2 immune receptors on NK and T cells [22], but also has the functional capacity to mediate virus-specific CTL responses during adaptive immunity. UL40-specific HLA-E-restricted (HLA-EUL40) CD8 T cells are only emerging as key players in HCMV immunity, and only a small amount of data are available on their phenotype and functions during the course of infection. We and others have recently reported on the long-term persistence of HLA-EUL40 CD8 T cells in the setting of organ transplantation, suggesting that this CTL population may belong and contribute to memory inflation and oligoclonal T cells expansion in HCMV-seropositive hosts [25,27,28,29,30]. The present study investigated the frequency and peptide specificity of long-lasting HCMV-specific CD8 T cells in a cohort of 120 seropositive healthy carriers with the aim to decipher the relative contribution of non-classical HLA-E- versus HLA-A2-specific CD8 T cells to long-term memory expansion and HCMV immunity.

## 2. Materials and Methods

### 2.1. Blood Samples and PBMC Isolation

This study included blood samples collected in 2020 from anonymous HCMV-seropositive (HCMV+) healthy blood donor volunteers (HV, *n* = 120). All blood donors were recruited at the French Blood Establishment (Etablissement Français du Sang des Pays de la Loire, Nantes, France, www.efs.sante.fr: URL accessed on 10 March 2023), a public institution under the responsibility of the French Ministry of Health, and informed consent was obtained from all individuals. According to French ethics laws, blood donation is based on voluntary participation, non-remuneration, anonymity and non-profitability. A declaration regarding the use of blood samples from blood volunteer donors for non-therapeutic research purpose was submitted to the French Ministry of Research, and the agreement number CPDL-PLER-2018 180 was assigned. All subjects were healthy and asymptomatic. Donors included 62 males and 58 females with a mean age value of 48.3 ± 14.9 years (median value: 49 years). Peripheral blood mononuclear cells (PBMC) were isolated by usual protocol using Ficoll Hypaque (Eurobio, Les Ulis, France) density gradient and kept frozen until use. PBMC were thawed in RPMI-1640 medium (Gibco, Amarillo, TX, USA) containing 10% human serum (pooled male AB sera, Sigma Aldrich, St-Louis, MI, USA), 2 mM L-glutamine (Gibco), 0.1 mg/mL streptomycin (Gibco) and 100 U/mL penicillin (Gibco).

### 2.2. MHC/Peptide Tetramer Complexes

Nine-mer peptides from the HCMV proteins UL40, UL83 (pp65) and UL123 (Immediate/Early1, IE1) were synthesized (purity >95%) by Genecust (Boynes, France). Three UL40 peptides from the leader peptide sequence (AA_15–23_: VMAPRTLIL, VMAPRTLLL and VMAPRSLLL) one UL83 (pp65) (AA_495–503_: NLVPMVATV) and one IE1 peptide (AA316-324: VLEETSVML) were produced and used to generate a set of five HLA/peptide monomers. The three UL40 peptides selected corresponded to the most frequently found in clinical samples and HCMV laboratory strains [29,31,32,33]. The HLA/peptide monomers used *HLA-E*01:01* and *HLA-A*02:01* alleles, and were produced by the recombinant protein core facilities (P2R, SFR Bonamy, Nantes Université, Nantes, France), as previously reported [29]. Recombinant HLA-A*02:01 proteins were produced with an alanine-to-valine mutation at AA position 245 to avoid non-specific binding of HLA-A2:01 molecules to CD8 molecules as reported previously [34]. Recombinant HLA proteins were produced in *E. coli* and refolded with 15 µg/mL of each UL40 peptide for HLA-E-monomers or pp65 and IE1 peptides for HLA-A*02:01-monomers. Next, HLA-monomers (HLA-E*01:01_UL40_ and HLA-A*02:01_pp65 or IE1_) were biotinylated for 4 h at 30 °C with BirA (6 µg/mL, Immunotech, Marseille, France), purified and tetramerized with APC-labeled streptavidin (BD Biosciences, Le Pont de Claix, France) as we described previously [29].

### 2.3. Flow Cytometry Protocol for the Detection and Quantification of HCMV-Specific CD8 T Cell Populations

An immunostaining protocol mixing monoclonal antibodies (mAb) and MHC/peptide tetramer complexes was set up for use on a BD FACS Canto™II cytometer (BD Biosciences, Fremont, CA, USA) with a 3-laser configuration (laser excitation wavelengths: 405 nm, 488 nm and 633 nm). As a preliminary step, titration experiments were carried out to establish the optimal mAb concentration providing the best staining index. After thawing, immunostaining was performed using a sequential multistep protocol. Before immunostaining, PBMC (1.10^6^ cells/well in 96-well plates) were washed twice in RPMI-1640 medium and filtered to eliminate cell clusters. PBMC were washed twice in PBS and centrifugated at 2500 rpm for 2 min at 4 °C. PBMC were incubated first with a blocking anti-CD94 mAb (clone HP-3D9, 5 µg/mL, BD Biosciences) in PBS/1%BSA (20 min, 4 °C) to prevent the binding of HLA-E_UL40_ tetramer complexes to CD94/NKG2 receptors in the following step. This blocking step ensure a specific recognition of TCR by HLA-E_UL40_ tetramer complexes without unspecific binding to CD94/NKG2 multimeric receptors as we reported previously [29,30]. PBMC were then incubated with one of the five APC-labeled HLA/peptide-tetramers (10 µg/mL, 1 h, 4 °C), before incubation (30 min, 4 °C) with the following mAbs: anti-CD3 (clone SK7/Leu4, APC-H7- labeled, BD Biosciences), anti-CD8α (clone SK1, BV510-labeled, BD Biosciences) and anti-γδTCR (clone B1, BV421-labeled, BD Biosciences). After 2 washing steps, PBMC were then incubated with a viability marker, DAPI (BD Biosciences), diluted in PBS/1%BSA (15 min, 4 °C). PBMC were washed twice in PBS before data acquisition. Immunostaining was performed in the presence of Fc-block™ reagent (BD Bioscience). For each sample, immunostaining with the different HLA/peptide complexes were performed in parallel in the same experiment. As negative controls, a Fluorescence Minus One (FMO) control, consisting of an immunostaining with all fluorochromes in the panel except one, was carried out for all mAbs and tetramers for each sample. Fluorescence compensations have been done with Ab-coated beads (anti-mouse Ig, κ chain/negative control compensation particles set, BD Biosciences). The frequency of pMHC tetramer-positive (Tet+) cell populations was determined using FlowJo™ Software v10 (BD Biosciences) based on manual gating strategies as reported on the results section. Relative CD4 and CD8 T cell percentages were determined using the proportion of live CD3+γδTCR- lymphocytes expressing or not the CD8 surface marker. The gating of cell populations and the data analysis strategy were identical for all samples. Using this protocol, our threshold of detection was established at 0.1% of total blood CD8+ T cells.

### 2.4. Data Analysis and Statistics

Results are expressed as means ± SD, or as medians ± interquartile range between Q1 and Q3, or percentages. Graphs and statistical analyses were carried out using GraphPad Prism^TM^ V7 Software (GraphPad, San Diego, CA, USA). For group comparisons, 2-sided paired Student’s *t* test with Mann–Whitney test or ANOVA were used as appropriate. A *p*-value *<* 0.05 was considered to be a statistically significant difference.

## 3. Results

### 3.1. Frequency and Pattern of HCMV-Specific CD8 T Cell Responses in HCMV+ Healthy Adults

A cohort of 120 successive HCMV+ healthy blood donors (HD) was investigated to detect and quantify the presence of HCMV-specific CD8 T cells using flow cytometry analysis. This analysis allowed us to define the frequency of HCMV+HD possessing anti-CMV CD8 T cells recognizing our set of HCMV peptides. As a result, we found that 60 out of the 120 HCMV+ HD (50%) possess at least one anti-HCMV CD8 T cell subset detected with our set of five pMHC class I tetramer complexes (Figure 1). Among the 120 HCMV+ HD, 30.8% display anti-HLA-A2pp65 CD8 T, 7.8% display anti-HLA-A2IE1 CD8 T and 27.5% display anti-HLA-EUL40 CD8 T (Figure 1A). HCMV+ HD with anti-HLA-A2pp65 CD8 T (*n* = 37) include HD with anti-HLA-A2pp65 CD8 T only (64.9%) or with a dual response composed of anti-HLA-A2pp65 and anti HLA-EUL40 CD8 T cells (35.1%). Concerning HD with anti-HLA-EUL40 CD8 T (*n* = 33), unique anti-HLA-EUL40 CD8 T responses were observed in 60.6% and in association with anti-HLA-A2pp65 for 39.4% of the HD (Figure 1B). Among HCMV+ hosts with anti-HLA-A2pp65 CD8 T cells 25.4% also possess anti-HLA-A2 IE1 CD8 T. Anti-HLA-A2IE1 CD8 T were not observed in the absence of anti-HLA-A2pp65 CD8 T responses. Concerning the HD with anti-HLA-EUL40 CD8 T responses (Figure 1C), in most of them (40.5%) CD8 T against the three HLA-E/peptides tested were detected while the hosts with CD8 T response against one and two HLA-E/peptide complexes account or 24.2% and 30.3%, respectively. VMAPRTLLL and VMAPRTLIL peptides were the targets for 84.8% of anti-HCMV CD8 T, while VMAPRSLLL account for 60.6% of the responses. VMAPRTLIL alone or in association with VMAPRSLLL and/or VMAPRTLLL are the most frequent patterns of HLA-EUL40 recognition. The various individual patterns of anti-HCMV T cell responses observed for the 60 HCMV+ HD are depicted in Figure 2.

### 3.2. Relative Frequencies of HCMV Peptide-Specific CD8 T Cell Responses among Total Blood CD8 T Cells and Impact on Host T Cell Homeostasis

The frequency of the anti-HCMV CD8 T cells was calculated and expressed as a percentage of total blood CD8 T cells for each host (Figure 3). Our gating strategy for flow cytometry analysis is depicted in Figure 3A, and was identical for all samples. Briefly, lymphocytes were gated on the basis of their morphology in FSC-A/SSC-A, and doublets of cells were excluded using SSC-A/SSC-H dot plots. Frequency of tetramers^+^ (Tet+) CD8α^+^T cell subpopulations was determined after gating on the CD3^+^ TCR γδ^−^ cells.

The percentages of anti-HCMV CD8 T cell subsets vary greatly among individuals. These variations were similar for the different CD8 T subsets within a range from 0.1 to 25.3% of total blood CD8 T cells (Figure 3B). The mean value (±SD) for the frequency of anti-HLA-A2pp65 CD8 T was 2.3 ± 3.3% and 2.5 ± 4.5% for anti-HLA-A2/IE1 CD8 T cells. HLA-E-restricted CD8 T responses represent 4.1 ± 5.3% of total CD8 T cells for the UL40 peptide VMAPRTLLL, 5.9 ± 7.5% for VMAPRTLIL and 3.5 ± 4.2% for VMAPRSLLL. The frequency of HLA-E/VMAPRTLIL was found to be significantly higher from the frequency of HLA-A2pp65 cells (*p* = 0.0354) suggesting that HLA-E-restricted CD8 T cells against the VMAPRTLIL UL40 peptide constitute a predominant subset among long-lasting anti-HCMV CD8 T cells.

Next, we sought to investigate whether the presence of HCMV-specific CD8 T cells in HCMV+ HD long time post infection could be a hallmark of an inflation of the CD8 T cell compartment. Based on previous studies from Cannon and colleagues [35], the average age of HCMV infection was 28.6 years in the US. Considering these data, we speculated that our cohort was representative of asymptomatic seropositive carriers far from primary acute infection and far from immune senescence, two conditions associated with inflated CD8 T cell pool. Figure 4A shows that the frequency of total blood CD8 T cells was around 35% of total T lymphocytes and was similar for both HCMV+ HD with or without HCMV-specific CD8 T cells detected in our experimental conditions. The mean value for CD4/CD8 ratio was found >1 for all the subjects, ranging from 0.42% to 4.69% for HD with detectable HCMV-specific CD8 T cells and from 0.41% to 5.85% for HD without detectable HCMV-specific CD8 T cells (Figure 4B), thus indicating no CD8 T inflation, but rather a normal homeostasis of T cells. Thus, the presence of detectable HCMV-specific CD8 T cells seems not to be associated with a particular increase in total CD8 T cells as also reflected by comparable CD4/CD8 ratio in both groups of seropositive individuals. Nevertheless, a further stratification of the hosts according to their age and to the time from infection to sampling would be necessary to refine these results.

### 3.3. Long-Lasting HCMV-Specific CD8 T Cells in Healthy Adults: TCR Avidity and Co-Receptor Expression

The use of pMHC multimers has been extensively utilized as the method of choice to analyze TCR avidity, especially for CD8+ T cells [36] and fluorescence intensity of tetramer binding may be use as a marker of TCR affinity. Therefore, in an attempt to compare the TCR avidity of the long-lasting CD8 T cells detected in the HCMV+ HD, the means of fluorescence intensity (MFI) of the different pMHC tetramer stainings were analyzed and compared (Figure 5). As reported Figure 5A, MFI vary greatly among HCMV+ HD hosts for the different CD8 T cell responses. Variations in MFI ranging from fold change increase x1 to 16 were found among the T cell responses, suggesting a large range of TCR avidity for the HCMV-specific CD8 T responses. Significantly higher MFI were observed for HLA-A2/IE1 CD8 T cells compared to HLA-A2/pp65 (*p* = 0.0457) and to HLA-E/VMAPRTLIL (*p* = 0.0374). A significant difference was also found between the MFI for HLA-A2/pp65 and HLA-E/VMAPRTLLL (*p* = 0.0422), with HLA-A2pp65 displaying the higher MFI. When several CD8 T cell responses are present concomitantly in hosts, different MFI are usually observed; the highest TCR avidity was observed for either HLA-A2 or HLA-E-restricted CD8 T subset, regardless (Figure 5B).

For the 60 hosts with HCMV-specific CD8 T response, expression for the TCR co-receptors CD3 and CD8 was measured and compared concomitantly for the pMHC Tet-negative (Tet-) blood CD8 T cells and Tet-positive (Tet+) for the five HCMV-specific CD8 T cell responses (Figure 5C,D). Concerning CD3 level, no significant differences were observed; only a trend toward a lower CD3 expression for HLA-A2/pp65 CD8 T compared to other CD8 subsets, which is consistent with previous studies [30]. No difference in CD8 expression was found, but a trend toward lower expression on HLA-A2/IE1 was. Together, these findings suggest that persistent memory HLA-E-restricted may display lower TCR avidity than HLA-A2-restricted T cell responses while exhibiting equal CD8 and equal or even higher CD3 surface expression.

## 4. Discussion

It is now well established that HCMV infection drives important and persistent changes in the homeostasis of immune cell subsets in hosts, such as promoting the expansion of highly differentiated populations of CD8 T, γδT and NK cells [8,9,10]. These changes may dramatically alter immunity against cancer and autoimmune diseases [17,37]. Under normal conditions, HLA-E allows the presentation of peptides derived from MHC class I leader sequences [38] to primarily regulate T cells [39] and NK cells [22] through the engagement of CD94/NKG2 receptors. Alternatively, HLA-E molecules can also present foreign peptides such as HCMV-derived peptides from the UL40 signal peptides. However, the potential role of T cells with specificity for HLA-E in immunity remains poorly understood. Deciphering the persistence of HLA-E-restricted CD8 T cells in HCMV-seropositive healthy individuals may help to understand their contribution to HCMV immunity and their bystander effects on other immune responses.

Using pMHC tetramers and conventional flow cytometry, here we report that 50% of HCMV-seropositive healthy individuals possess HCMV-specific blood CD8 T cells representing ≥0.1% of total circulating CD8 T cells years post-infection. Around a third (30.8%) of individuals possess a conventional HCMV-specific CD8 T cell response restricted by HLA-A (i.e., A2pp65 or A2IE1) and an equal proportion (27.5%) possess a non-conventional response directed against UL40 and restricted by the non-classical MHC molecules HLA-E. Thus, a major finding from this study was to establish an equivalent long-term prevalence for both, conventional and unconventional, types of responses. In this study, the HLA genotyping was not performed on the HCMV+ blood donors. Nevertheless, the frequency of HCMV+ HD with HLA-A2pp65 CD8 T cell response that we observed was identical (30.8%) to the HLA-A*02:01 allele frequency reported in our local population (30.8% from *n* = 42,623 individuals in the Nantes area, France, http://www.allelefrequencies.net, URL (accessed on 13 February 2023)). This may suggest that in fact all HLA-A02:01 carriers from our cohort display HLA-A2pp65 CD8 T cells. Only a quarter (25.8%) of HCMV+ HD with an HLA-A2pp65 response also possess a HLA-A2IE1 response and no HLA-A2IE1 T cells was observed in the absence of HLA-A2pp65 T cells. These results may suggest a higher prevalence of long-lasting CD8 T cells targeting pp65 than IE1 antigens.

A concomitant detection of A2pp65 and EUL40 was found for a third of HD with an A2pp65 T response and vice versa. This result is consistent with our previous study, showing that EUL40 CD8 T cells were frequently found in HLA-A2 virus carriers in a cohort of kidney transplant recipients [29]. Consistent with this association, the signal peptide sequence provided by the HLA-A*02:01 alleles, VMAPRTLVL, is distinct from the UL40 peptide variants used here (VMAPRTLLL, VMAPRTLIL and VMAPRSLLL), allowing HLA-A*02:01 carriers to develop CD8 T responses against these UL40 peptides recognized as non-self peptides. Thus, these responses appear limited to individuals who lack HLA-I alleles that encode these sequences, presumably reflecting tolerance to the HLA-I-encoded peptides.

The UL40_15–23_ peptide VMAPRTLIL was found to be the predominant target of EUL40 CD8 T responses in our cohort. Consistent with this result, the distribution of HLA-E binding UL40_15–23_ peptides was analyzed and compared to those of HLA Class I observed in a cohort of 444 healthy individuals also reported the VMAPRTLIL peptide as the most prevalent peptide in UL40 and HLA-C [40]. The VMAPRTLIL peptide binds HLA-E and interact with both CD94/NKG2A and CD94/NKG2C with high affinity [32]. CD8 T reactive to HLA-E_VMAPRTLIL_ were most often found associated with HLA-E_VMAPRTLLL_ and/or HLA-E_VMAPRSLLL_ responses.

We should acknowledge that our attempt to establish a qualitative and quantitative profile of persistent conventional, HLA-A2-reactive, versus unconventional, HLA-E-reactive, CD8 T cells suffers some limitations. First, we cannot exclude that HCMV-specific CD8 T cells representing less than 0.1% of the total CD8 T pool were also present this cohort but were not detected in our experiments. Moreover, concerning conventional responses, we focused on pp65 and IE1, but other epitopes were not tested here. Second, concerning HLA-E-reactive CD8 T cells, here we used three HLA-EUL40 complexes corresponding to the most frequent UL40_15–23_ sequence variants, but we cannot exclude the possibility that CD8 T cells targeting other UL40 variants were also present this cohort but were not detected using our protocol.

Different parameters affect the sensitivity that T cells will display against the pMHC. TCR affinity defines the strength of the interaction between a unique TCR and pMHC and is usually determined by surface plasmon resonance. In another way, TCR avidity depicts the contact of multiple TCRs and pMHCs [36]. Consequently, multimers of fluorochrome-labeled pMHC are commonly used to stain peptide-specific T cells and to measure their TCR avidity [36]. This parameter also takes into account the effect of T-cell co-receptors such as CD3 and CD8 in the stabilization of TCR–pMHC binding [41]. Here we showed that HCMV-reactive CD8 T cells display a large diversity of TCR avidity which vary among hosts and T cell responses. Our results may suggest that HLA-EUL40 CD8 T cells have lower TCR avidity than A2pp65 or A2IE1 CD8 T cells. Similar to TCR avidity, a high degree of heterogeneity was observed but no significant change was found for the expression of CD3 and CD8 on the different CD8 T subsets. Nevertheless, irrespective of the level of CD8 surface expression, it has been shown that HLA-E bound CD8αα with an affinity far lower than the majority of classical HLA alleles, including HLA*02:01, due to amino acid differences adjacent to the HLA α3 domain loop [42]. Moreover, pMHC tetramers do not provide information on functional avidity [43]. T cell activation and effector functions, namely, T-cell proliferation, cytotoxic activity, cytokine production have been shown to vary greatly among HCMV-specific CD8 T cells [44].

Functionally, the role that HLA-EUL40 CD8 T cells may play in HCMV maintenance, latency or, conversely, in virus clearance is still unknown. In particular, whether HLA-EUL40 CTL could efficiently contribute to the immune control of HCMV infection, as reported for conventional HLA-A2pp65 CD8 T cells, still requires investigation. If proven, this could offer the possibility of using HLA-EUL40 CD8 T cells as diagnostic tools through the development of a routine assay to detect these T cell subset in patients. Such application would require, as a preliminary step, accurate monitoring of HLA-EUL40 CD8 T cells in hosts along HCMV infection according to age, gender and immune status. In contrast to the highly polymorphic HLA class Ia genes, the *HLA-E* gene display a low polymorphism. Two major alleles HLA-E*01:01 and HLA-E*01:03 have been reported [45], and both are equally distributed in most populations. Consequently, *HLA-E* encodes for two HLA-E proteins that differ by 1 amino acid in position 107 (HLA-E*01:01/Arg107, HLA-E*01:03/Gly107). This may help to develop a routine, «universal», with no HLA restriction, assay for the detection of these T cells in hosts to follow immune reconstitution in HSC recipients for instance. Moreover, the efficacy of HLA-EUL40 CTL to control the infection may also provide a useful source of cells with low or even no HLA-restriction with high potential for immune cell therapy such as adoptive transfer, as proposed to prevent and cure HCMV severe infections [46]. On the other hand, the bystander role that the long-term persistence of HLA-EUL40 CD8 T cell populations induced in response to the common HCMV infection also requires attention. Vascular cells [47] and cancer cells [48] express high level of HLA-E presenting self peptides from HLA Ia leader peptides, and thus may be targeted by HLA-EUL40 CD8 T cells due to peptide sequence mimicry. The consequences for allograft rejection, vascular complications in HCMV infection and tumor progression are only emerging [48].

## 5. Conclusions

The present investigation further refined the frequency and peptide specificity of long-lasting HCMV-specific CD8 T cells in a cohort of 120 seropositive HCMV+ healthy carriers. Our findings established that unconventional HLA-E-restricted UL40-reactive CD8 T cells were almost as frequent as the “immunodominant” conventional HLA-A2-restricted pp65- and IE1-specific CD8 T responses years or even decades post infection. Under our experimental conditions, we showed that around 30% of HCMV+ healthy individuals possess HLA-A2-restricted (A2pp65 or A2IE1), and an equal proportion possess HLA-EUL40 HCMV-specific CD8 T cells, representing ≥0.1% of total blood CD8 T cells. The frequency of CD8T cells against the UL40 peptide VMAPRTLIL presented by HLA-E molecules among was significantly higher, among total blood CD8 T cells, than the frequency of HLA-A2pp65 T cells. Thus, HLA-EUL40 CD8 T cell responses represent a large and stable pool of HCMV-specific CD8 T cells among circulating CD8+ T cells that are maintained for life. Functionally, HLA-EUL40 CD8 T cells display lower TCR avidity, but similar levels of CD3 and CD8 co-receptors compared to conventional HLA-A2pp65 T cells. Our data indicate that HLA-E-restricted CD8 T cells against the VMAPRTLIL UL40 peptide constitute a predominant subset among long-lasting anti-HCMV CD8 T cells. These findings further support unconventional HLA-E-restricted CD8 T cells as a major trigger of HCMV immunity. While the phenotypical and functional features of HLA-EUL40-specific CD8 T cells started to be elucidated [24,25,26,30,49], their clinical significance still remains to be defined.

## Figures and Tables

**Figure 1 cells-12-00889-f001:**
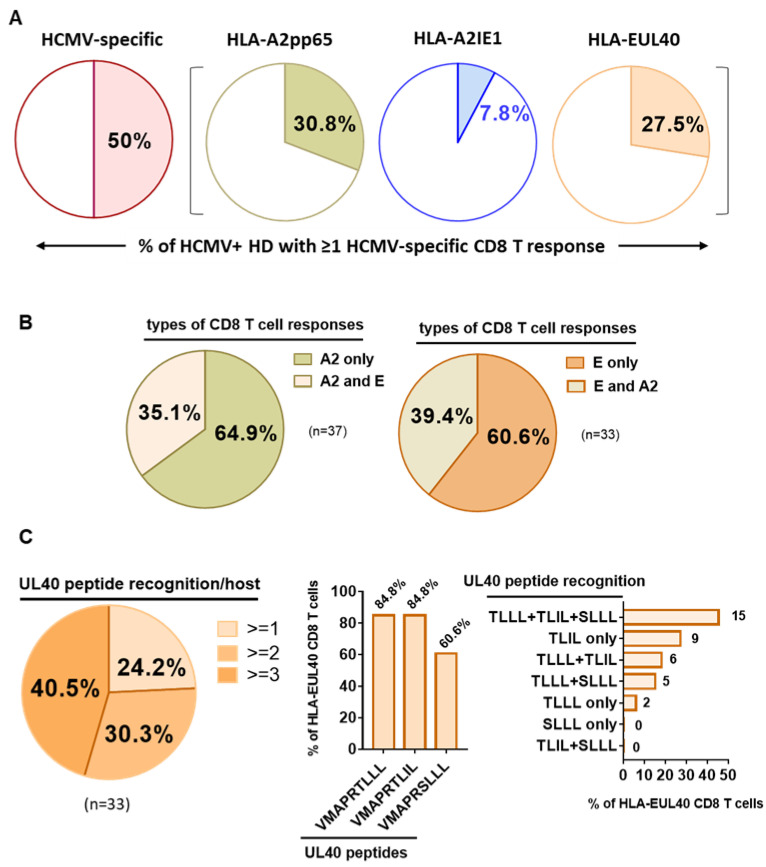
Detection and frequency of HLA-A2pp65, HLA-A2IE1 and HLA-EUL40 HCMV-specific CD8 T cells in HCMV+ HD. Pie charts indicating the percentages of HCMV+ HD with HCMV-specific CD8 T cells detected. A single PBMC sample per donor was used for the concomitant detection of HLA-A2pp65, HLA-A2IE1 and HLA-EUL40 CD8 T cells. For the detection of HLA-EUL40 CD8 T cells, three HLA-E tetramers loaded with three different peptides (VMAPRTLLL, VMAPRTLIL and VMAPRSLLL) corresponding the most frequent UL40 variants were used in parallel experiments. (**A**) Percent of hosts (*n* = 120) with at least one type of CD8 T cell response (all, A2pp65, A2IE1 or EUL40). (**B**) Percent of hosts with only an A2pp65 response or with both A2pp65 and EUL40 CD8 T cell responses among hosts with A2pp65 response (*n* = 37, left panel) and with only an EUL40 response or with both A2pp65 and EUL40 CD8 T cell responses among hosts with EUL405 response (*n* = 33, right panel). (**C**) Analysis of HLA-EUL40 CD8 T responses (*n* = 33) in hosts showing the number (left panel), sequences (medium panel) and combinations (right panel) of UL40 peptide recognition.

**Figure 2 cells-12-00889-f002:**
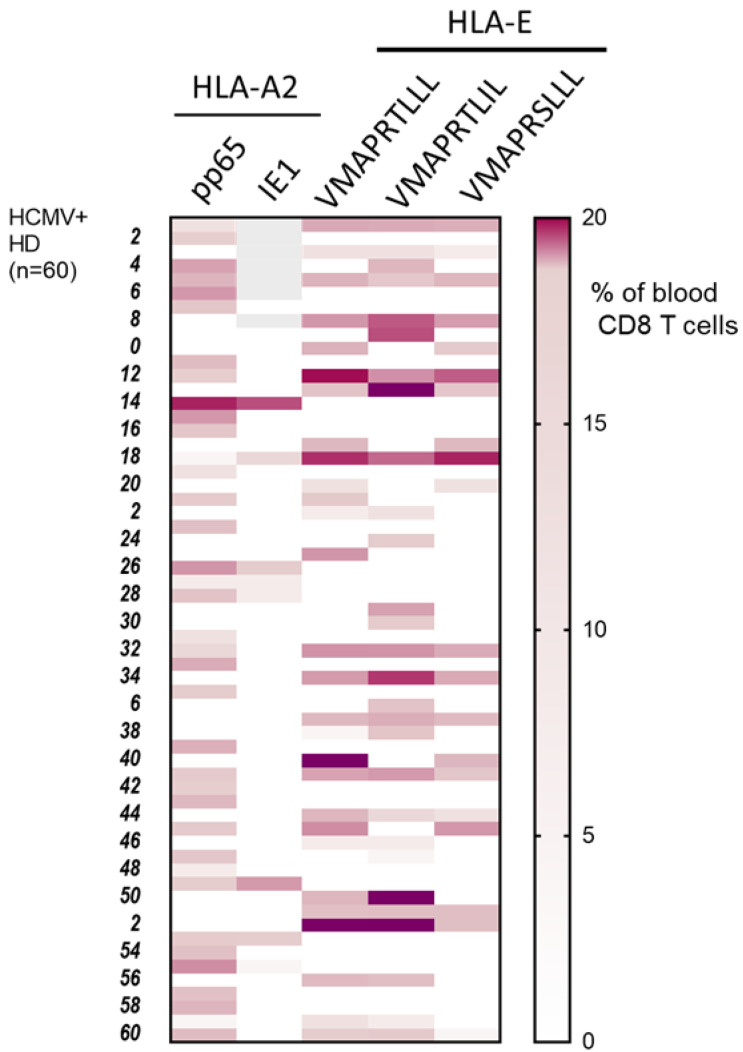
Patterns of long-lasting HCMV-specific CD8 T cell subsets detected in HCMV+ adults. Heat map shows the presence and the specificity of the HCMV-specific CD8 T cells detected for 60 HCMV+ HD out of the cohort study. HCMV-specific CD8 T cell subsets were detected and quantified by flow cytometry after immunostaining using a set of pMHC class I tetramers containing HLA-A2pp65, HLA-A2IE1 and HLA-EUL40 (HLA-E/VMAPRTLLL, HLA-E/VMAPRTLIL, HLA-E/VMAPRSLLL) tetramers. Immunostainings for the five pMHC tetramers were performed in parallel during the same experiment for each PBMC sample. Results display the percentages of the HCMV-specific CD8 T subsets among total blood CD8 T cells using a color scale.

**Figure 3 cells-12-00889-f003:**
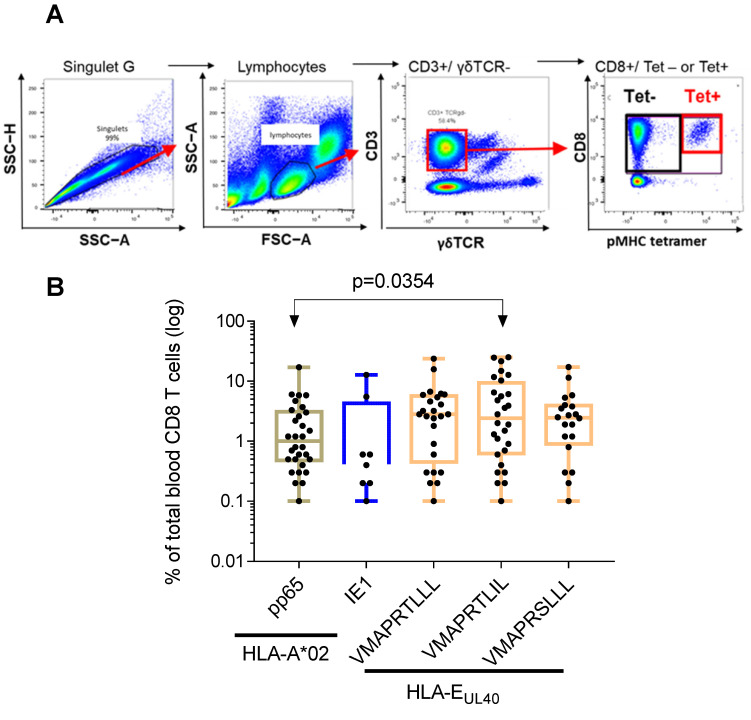
Frequency distribution of HCMV-specific CD8 T cells among total blood CD8 T cells in healthy HCMV+ adults. (**A**) Gating strategy for flow cytometry analysis. Serial steps of cell selection to identify and quantify pMHC tetramer positive (Tet+) are indicated. (**B**) Scatter plots reporting the frequency of A2/pp65, A2/IE1 and E/UL40 CD8 T cells in blood samples from HCMV+ HD with at least one CD8 T response (*n* = 60). Data are expressed as percent of total blood CD8 T cells for each HD. Medians and IQR are shown. Each dot corresponds to a single, independent, CD8 T cell subset. Statistical analysis was performed by Mann–Whitney *U*-test.

**Figure 4 cells-12-00889-f004:**
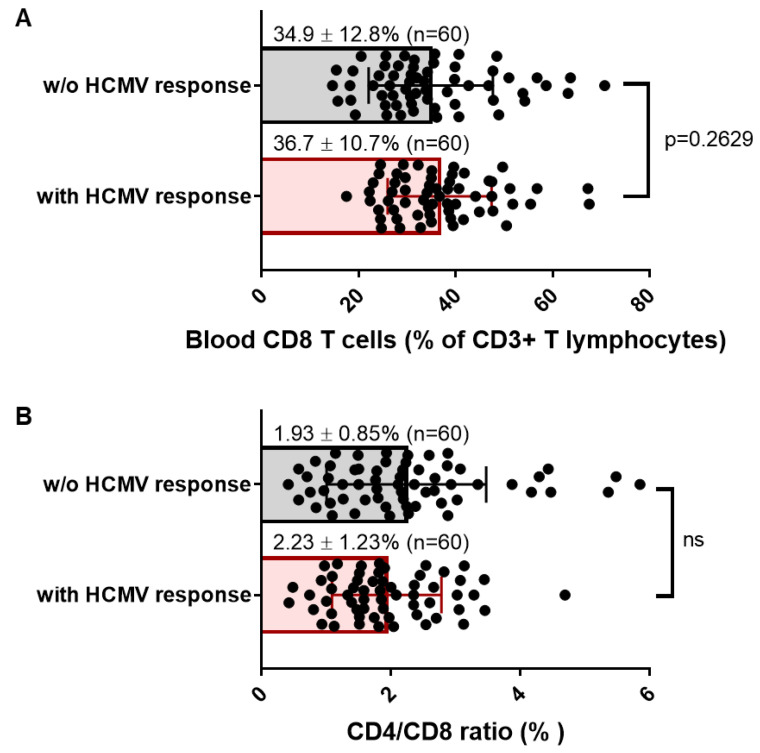
Frequency of blood CD8 and CD4/CD8 T cell ratio in HCMV+ healthy adults with or without detectable HCMV-specific CD8 T cells. (**A**) Scatter plots reporting the frequency of total CD8 T cells in blood samples from HCMV+ HD with at least one CD8 T response (*n* = 60) or with no response detected under our conditions (*n* = 60). Data are expressed as percent of total blood CD8 T cells for each HD. (**B**) Scatter plots reporting the CD4T/CD8 T cell ratio in blood samples from HCMV+ HD with at least one CD8 T response (*n* = 60), or with no response detected under our conditions (*n* = 60). Means and SD are shown. Each dot corresponds to a single PBMC sample from an individual HD. Statistical analysis was performed by Mann–Whitney *U*-test; *p* value is indicated.

**Figure 5 cells-12-00889-f005:**
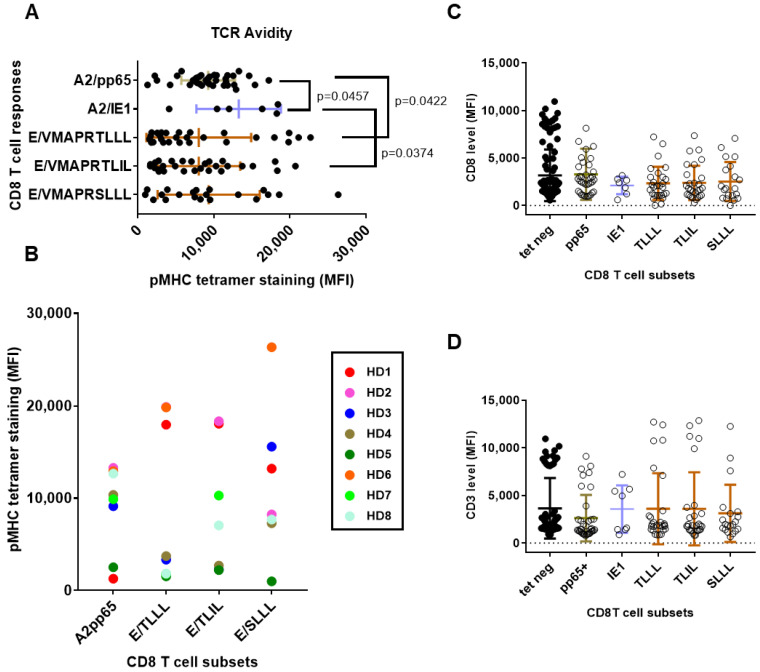
Characteristics of long-lasting HCMV-specific CD8 T cell responses according to HLA allele and peptide specificity: TCR avidity and co-receptor expression. (**A**) Scatter plots reporting the means of fluorescence intensity (MFI) for pMHC tetramer staining in the five subsets of HCMV-specific CD8 T cells. (**B**) Patterns of TCR avidity for HCMV-specific CD8 T cells according to HLA allele and peptide specificities. Relative pMHC tetramer staining for HLA-A2- versus HLA-E-reactive CD8 T cells for eight HCMV+ HD (HD1-HD8) with multiple CD8 T cell responses are reported on the graph. Each color corresponds to an individual HCMV+ HD. (**C**,**D**) Scatter plots reporting the MFI for CD3 (**C**) and CD8 (**D**) on the five subsets of HCMV-specific CD8 T cells and in pMHC tetramer-negative (tet neg) CD8 T cells from the same hosts. Each dot corresponds to a single, independent, CD8 T cell subset. Statistical analysis was performed by Mann–Whitney *U*-test; *p* values are indicated. TLLL:VMAPRTLLL, TLIL:VMAPRTLIL, SLLL:VMAPRSLLL.

## Data Availability

The data presented in this study are available on request from the corresponding author.

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
