# Peer review of "Persistent CD8 T Cell Marks Caused by the HCMV Infection in Seropositive Adults: Prevalence of HLA-E-Reactive CD8 T Cells"

_cells, 2023, doi:10.3390/cells12060889_

Round 1
Reviewer 1 Report
This study investigated the frequency and selected peptide specificity of long lasting HCMV-specific CD8 T cells in a cohort of 120 seropositive (HCMV+) healthy carriers with the aim to decipher the relative contribution of unconventional HLA-E- versus conventional HLA-A2-specific CD8 T cells to long-term T cell memory expansion in HCMV immunity. This was assessed via tetramer and antibody labeling of T cells in flow cytometry. They found that HLA-20 EUL40 CD8 T cells display lower TCR avidity but similar levels of CD3 and CD8 coreceptors. In conclusion, HLA-E-restricted CD8 T cells against the VMAPRTLIL UL40 peptide constitute a predominant subset among long lasting anti-HCMV CD8 T cells.
Overall, the paper does a nice job of presenting data and experiments are rigorous. The authors also do a nice job discussing their data and mentioning some limitations, with the exceptions of what is listed below.
The introduction stresses the importance of understanding the phenomenon of memory inflation, i.e. the oligoclonal expansion of HCMV-specific T cells that contribute to immune dysregulation in the elderly population. But the age group median of 49 years old for blood analysis does not correlate with the population that would be suffering from memory inflation. And Figure 4 indicates that the HCMV+ population does not have T cell inflation compared to those who were not infected. Perhaps this is not the correct age group to study to make conclusions on this aspect.
This manuscript has the aim to decipher the relative contribution of nonclassical HLA-E- versus HLA-A2-specific CD8 T cells to long-term memory expansion and HCMV immunity as stated at the end of the introduction. However, there is no explanation of how they know the sampled individuals have not recently overcome HCMV infection, vs those that did it decades ago. There is no discussion or data shown that address how the difference in time from infection could affect results. Perhaps the differences they see in individuals in immunodominant T cell recognition is entirely dependent on how long ago they were infected. The authors need to make clear how they know they are looking at responses that are decades old as stated, and not just several months or a few years.
The attributes of HCMV-specific T cell subsets is not clearly related to any subset of patients via age, sex, immune status, or time from primary infection, so it is difficult to understand what any of the clinical implications might be for the T cell population frequencies that have been identified. Relating observations to particular subsets of patients may be very helpful in development of diagnostic tools/clinical markers as discussed.
Author Response
Point by point answers to the reviewer’s comments
Reviewer 1
This study investigated the frequency and selected peptide specificity of long lasting HCMV-specific CD8 T cells in a cohort of 120 seropositive (HCMV+) healthy carriers with the aim to decipher the relative contribution of unconventional HLA-E- versus conventional HLA-A2-specific CD8 T cells to long-term T cell memory expansion in HCMV immunity. This was assessed via tetramer and antibody labeling of T cells in flow cytometry. They found that HLA-20 EUL40 CD8 T cells display lower TCR avidity but similar levels of CD3 and CD8 coreceptors. In conclusion, HLA-E-restricted CD8 T cells against the VMAPRTLIL UL40 peptide constitute a predominant subset among long lasting anti-HCMV CD8 T cells.
Overall, the paper does a nice job of presenting data and experiments are rigorous. The authors also do a nice job discussing their data and mentioning some limitations, with the exceptions of what is listed below.
The introduction stresses the importance of understanding the phenomenon of memory inflation, i.e. the oligoclonal expansion of HCMV-specific T cells that contribute to immune dysregulation in the elderly population. But the age group median of 49 years old for blood analysis does not correlate with the population that would be suffering from memory inflation. And Figure 4 indicates that the HCMV+ population does not have T cell inflation compared to those who were not infected. Perhaps this is not the correct age group to study to make conclusions on this aspect.
- RE. We thank the reviewer for these remarks. We agree with your comment concerning on the correlation between the age and the occurence of memory inflation. Our aim in the present study was the to investigate the frequence and the relative proportion of « conventional » versus « unconventional’ HCMV-specific CD8 T cells responses in seropositive adults with during viral latency, probably years post infection, but before memory inflation and immune senescence as reported in ederly. Our study includes only HCMV seropositive (HCMV+) subjects. Consequently Figure 4 shows the percentages of CD8 T cells and the CD4/CD8 ratio for HCMV+ adults with or without detectable HCMV- specific CD8 T cells. From our point of view the lack of significant difference between the two groups confirm the fact that samples are far from acute infection and also far from immune senescence, two conditions which should be reflected by an expanded CD8 T compartment. Nevertheless, we acknowledge that longitudinal time course studies exploring the evolution of anti-HCMV CD8 T cell responses post infection would be very interesting for our understanding of immune senescence and aging.
This manuscript has the aim to decipher the relative contribution of nonclassical HLA-E- versus HLA-A2-specific CD8 T cells to long-term memory expansion and HCMV immunity as stated at the end of the introduction. However, there is no explanation of how they know the sampled individuals have not recently overcome HCMV infection, vs those that did it decades ago. There is no discussion or data shown that address how the difference in time from infection could affect results. Perhaps the differences they see in individuals in immunodominant T cell recognition is entirely dependent on how long ago they were infected. The authors need to make clear how they know they are looking at responses that are decades old as stated, and not just several months or a few years.
Re : We are grateful to the reviewer for pointing this out. Based on previous studies from Cannon and colleagues, The average age of HCMV infection was 28.6 years in US (Colugnati et al, 2007, doi: 10.1186/1471-2334-7-71). Considering these data we speculate that our cohort was representative of seropositive asymptomatic healthy carriers far from primary acute infection. As suggested our manuscript has been modified to answer these concerns and to better introduced the samples used.
The attributes of HCMV-specific T cell subsets is not clearly related to any subset of patients via age, sex, immune status, or time from primary infection, so it is difficult to understand what any of the clinical implications might be for the T cell population frequencies that have been identified. Relating observations to particular subsets of patients may be very helpful in development of diagnostic tools/clinical markers as discussed.
RE : We are grateful to the reviewer for pointing this out and we included this limitation in the discussion.
Reviewer 2 Report
This well written paper by Rousellière and co-workers analyzes the frequencies of CMV specific T cells in healthy CMV-positive blood donors. They use tetramer staining against known HLA2 and -E epitopes and enumeration by flow cytometry. This is a well established and stringent approach. The analysis was carried out in >100 donors, providing robust data on prevalence of HLA-E-reactive CD8 T cells.
Two points are critical to me:
1. Data one the functional and7or phenotypic characterization of these cells are missing. This would nicely complement the presented data on tetramer avidity.
2. The authors discuss the role of these cells for memory inflation (line 240ff.) and conclude, that there is no contribution. This asumption is based on the comparison of donors with or without a response against the known A2-restricted epitopes for pp65 and IE-1. I think this interpretation is not correct. Although IE-1 and pp65 are immmunodominat proteins, still a relevant proportion of individuals are not responding (see paper from Sylwester et al., 2005). The method used does not provide an accurate estimate of the HCMV-specific response in a donor (which could be directed against other epitopes or proteins).
Author Response
This well written paper by Rousellière and co-workers analyzes the frequencies of CMV specific T cells in healthy CMV-positive blood donors. They use tetramer staining against known HLA2 and -E epitopes and enumeration by flow cytometry. This is a well established and stringent approach. The analysis was carried out in >100 donors, providing robust data on prevalence of HLA-E-reactive CD8 T cells.
Two points are critical to me:
- Data one the functional and7or phenotypic characterization of these cells are missing. This would nicely complement the presented data on tetramer avidity.
RE : Thank you for this comment. We agree that a better understanding of the role of this unconventional CD8 T cell subset is a major point to resolve. In particular whether they can play regulatory functions toward other immune cells expressing HLA-E and thus participate to the homeostasis of the immune responses is of importance. In our previous studies (Jouand N. et al. Plos Pathogens, 2018 ; Rousselière A. Front. Immunol., 2022) we provided comparative analyses on the phenotype of HLA-EUL40 versus HLA-A2 CD8T cells. We reported that both HLA-EUL40 and HLA-A2pp65 CD8T display a phenotype specific of CD8+ TEMRA (CD45RA+/CCR7-) but HLA-EUL40 CD8T express distinctive level for CD3, CD8 and CD45RA. Tim3, Lag-3, 4-1BB, and to a lesser extend 2B4 are hallmarks for T cell priming post-primary infection while KLRG1 and Tigit are markers for restimulated and long lived HCMV-specific CD8T responses. These cell markers are equally expressed on HLA-EUL40 and HLA-A2pp65 CD8T. In contrast, CD56 and PD-1 are cell markers discriminating memory HLA-E- from HLA-A2-restricted CD8T. This phenotype was found stable and identical in healthy HCMV seropositive individuals and in kidney transplant recipents after a primary infection or a HCMV reactivation. Functionnally, we found that long lived HLA-EUL40 are CTL displaying higher proliferation rate compared to HLA-A2pp65 CD8T consistent with elevated CD57 expression.
- The authors discuss the role of these cells for memory inflation (line 240ff.) and conclude, that there is no contribution. This asumption is based on the comparison of donors with or without a response against the known A2-restricted epitopes for pp65 and IE-1. I think this interpretation is not correct. Although IE-1 and pp65 are immmunodominat proteins, still a relevant proportion of individuals are not responding (see paper from Sylwester et al., 2005). The method used does not provide an accurate estimate of the HCMV-specific response in a donor (which could be directed against other epitopes or proteins).
RE : We thank the reviewer for this pertinent comment. We fully agree with this concern and indeed we cannot conclude from our data that anti-HCMV CD8T cells (either HLA-A2- or/and HLA-E-specific) have no role in memory inflation. We can only say that the presence of large amount of HCMV-specific CD8 T cells is not associated with a larger pool of CD8T cells. The text has been modified to avoid such misinterpretation and to clarify this point. The technical limitation of the present analysis has also been introduced to modulate the interpretation as you suggested.